# SimO Loss: Anchor-Free Contrastive Loss for Fine-Grained Supervised Contrastive Learning

## Abstract

We introduce a novel anchor-free contrastive learning (AFCL) method leveraging our proposed Similarity-Orthogonality (SimO) loss. Our approach minimizes a semi-metric discriminative loss function that simultaneously optimizes two key objectives: reducing the distance and orthogonality between embeddings of similar inputs while maximizing these metrics for dissimilar inputs, facilitating more fine-grained contrastive learning. The AFCL method, powered by SimO loss, creates a fiber bundle topological structure in the embedding space, forming class-specific, internally cohesive yet orthogonal neighborhoods. We validate the efficacy of our method on the CIFAR-10 dataset, providing visualizations that demonstrate the impact of SimO loss on the embedding space. Our results illustrate the formation of distinct, orthogonal class neighborhoods, showcasing the method's ability to create well-structured embeddings that balance class separation with intra-class variability. This work opens new avenues for understanding and leveraging the geometric properties of learned representations in various machine-learning tasks.

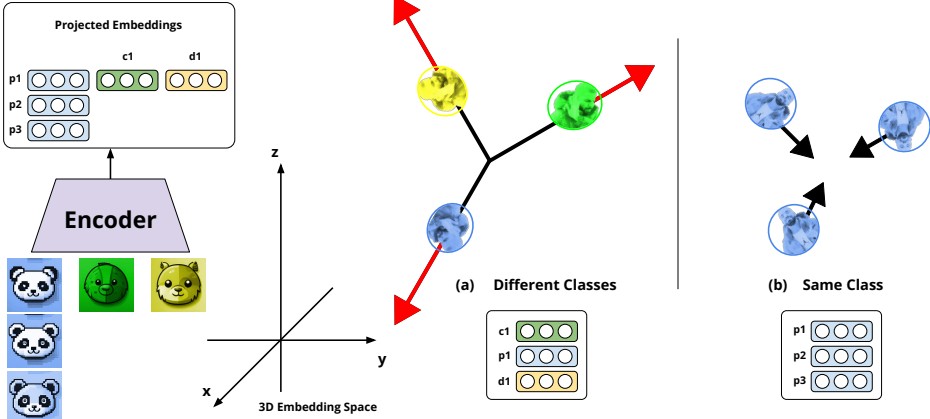

Figure 1: 3D interpretation of Anchor-Free Contrastive Learning (AFCL) using Similarity-Orthogonality (SimO) loss: (a) On the left, all the samples are negatively contrasted with each other. The loss function aims to push them away from each other while maintaining the orthogonality between all the embedding vectors in our embedding Space. (b) On the right, all the data points belong to the same class. The loss function here decreases the orthogonality and the distance between embeddings of the same class.

## 1 Introduction

The pursuit of effective representation learning (Gidaris et al. (2018); Wu et al. (2018); Oord et al. (2019)) has been a cornerstone of modern machine learning, with contrastive methods emerging as particularly powerful tools in recent years. Despite significant advancements,

the field of supervised contrastive learning (Khosla et al. (2021); Balestriero et al. (2023)) continues to grapple with fundamental challenges that impede the development of truly robust and interpretable models.

Our research unveils persistent challenges in embedding methods (Wen et al. (2016); Grill et al. (2020); Hjelm et al. (2019)), notably the lack of interpretability and inefficient utilization of embedding spaces due to dimensionality collapse (Zbontar et al. (2021); Jing et al. (2022)). Certain techniques, such as max operations in loss functions (e.g., max(0, loss)) (Gutmann & Hyvärinen (2010)) and triplet-based methods (Sohn (2016b); Tian et al. (2021)), introduce complications like non-smoothness, which disrupt gradient flow. These approaches also suffer from biases in hand-crafted triplet selection and necessitate extensive hyperparameter tuning, thereby limiting their generalizability (Rusak et al. (2024)). Contrastive loss functions, including InfoNCE (Chen et al. (2020)), often rely on large batch sizes and negative sampling, leading to increased computational costs and instability. While recent metric learning approaches (Movshovitz-Attias et al. (2017); Qian et al. (2020)) have made strides in improving efficiency and scalability, they frequently sacrifice interpretability. This trade-off results in black-box models that offer minimal insight into the learned embedding structure.

We present SimO loss, a novel AFCL framework that addresses these long-standing issues. SimO introduces a paradigm shift in how we conceptualize and optimize embedding spaces. At its core lies a carefully crafted loss function that simultaneously optimizes Euclidean distances and orthogonality between embeddings – a departure from conventional approaches that typically focus on one or the other (Schroff et al. (2015)).

The key innovation of SimO is its ability to project each class into a distinct neighborhood that maintains orthogonality with respect to other class neighborhoods. This property not only enhances the explainability of the resulting embeddings but also naturally mitigates dimensionality collapse by encouraging full utilization of the embedding space. Crucially, SimO operates in a semi-metric space, a choice that allows for more flexible representations while preserving essential distance properties.

From a theoretical standpoint, SimO induces a rich topological structure in the embedding space, seamlessly blending aspects of metric spaces, manifolds, and stratified spaces. This unique structure facilitates efficient class separation while preserving nuanced intra-class relationships – a balance that has proven elusive in previous work. The semi-metric nature of our approach, allowing for controlled violations of the triangle inequality, enables more faithful representations of complex data distributions that often defy strict metric assumptions.

Our key contributions are:

- We propose an anchor-free pertaining method (AFCL) for supervised and semi-supervised contrastive learning
- We introduce SimO, an anchor-free contrastive learning loss that significantly advances the state-of-the-art in terms of embedding explainability and robustness.
- We provide a comprehensive theoretical analysis of the induced semi-metric embedding space, offering new insights into the topological properties of learned representations.

As we present SimO to the community, we do so with the conviction that it represents not just an incremental advance, but a fundamental reimagining of contrastive learning—one that addresses the core challenges that have long hindered progress in the field.

## 2 RELATED WORK

### 2.1 ANCHOR-BASED CONTRASTIVE LEARNING

In contrastive learning, anchor-based losses have evolved from simple pairwise comparisons to more sophisticated multi-sample approaches (Khosla et al. (2021)). The triplet loss (Co-

ria et al. (2020)), which compares an anchor with one positive and one negative sample, has found success in applications like face recognition (Chopra et al. (2005)), despite its tendency towards slow convergence. Building on this foundation, the (N+1)-tuplet loss extends the concept to multiple negatives, approximating the ideal case of comparing against all classes. Further refinement led to the development of the multi-class N-pair loss, which significantly improves computational efficiency through strategic batch construction, requiring only 2N examples for N distinct (N+1)-tuplets (Sohn (2016a)). Recent theoretical work has illuminated the connections between these various loss functions. Notably, the triplet loss can be understood as a special case of the more general contrastive loss. Moreover, the supervised contrastive loss (Khosla et al. (2021)), when utilizing multiple negatives, bears a close resemblance to the N-pairs loss. Nevertheless, anchor-based methods are sensitive to negative sample quality, which can lead to inefficiencies in small datasets and struggle with false negatives. It also relies heavily on effective data augmentations and large batch size, with a risk of overlooking global relationships.

## 2.2 DIMENSIONALITY COLLAPSE IN CONTRASTIVE LEARNING METHODS

Dimensionality collapse, a significant challenge in contrastive learning, occurs when learned representations converge to a lower-dimensional subspace, thereby diminishing their discriminative power and compromising the model's ability to capture data structure effectively (Jing & Tian (2020)). To address this issue, researchers have proposed several innovative strategies. The NT-Xent loss function (Chen et al. (2020)) implements temperature scaling to emphasize hard negatives, promoting more discriminative representations . Another approach involves the use of a nonlinear projection head, which enhances representation quality through improved hypersphere mapping (Grill et al. (2020)). The Barlow Twins method (Zbontar et al. (2021)) takes a different tack, focusing on redundancy reduction by minimizing correlations between embedding vector components through optimization of the cross-correlation matrix. Architectural innovations have also played a crucial role in combating dimensionality collapse. Methods like BYOL and SimSiam employ asymmetric architectures to prevent the model from converging to trivial solutions (Chen & He (2021)). The use of stop gradient in these methods ensures that the models do not converge to produce the same outputs over time. Additionally, the use of batch normalization (Ioffe (2015)) has been empirically shown to stabilize training and prevent such trivial convergence, although the precise mechanisms underlying its effectiveness remain an area of active research (Peng et al. (2023)).

## 2.3 EXPLAINABILITY OF CONTRASTIVE LEARNING

The underlying mechanisms driving the effectiveness of contrastive learning remain an active area of investigation. To shed light on the learned representations, Zhu et al. (2021) introduced attribution techniques for visualizing salient features. Cosentino et al. (2022) explored the geometric properties of self-supervised contrastive methods. They discovered a non-trivial relationship between the encoder and the projector, and the strength of data augmentation with increasing complexity. They provided a theoretical framework for understanding how these methods learn invariant representations based on the geometry of the data manifold. Furthermore, Steck et al. (2024) examined the implications of cosine similarity in embeddings, challenging the notion that it purely reflects similarity and suggesting that its geometric properties may influence representation learning outcomes. Wang & Liu (2021) investigate the behavior of unsupervised contrastive loss, highlighting its hardness-aware nature and how temperature influences the treatment of hard negatives. They show that while uniformity in feature space aids separability, excessive uniformity can harm semantic structure by pushing semantically similar instances apart. Wang & Isola (2020) identified alignment and uniformity as key properties of contrastive learning. Alignment encourages closeness between positive pairs, while uniformity ensures the even spread of representations on the hypersphere. Their work demonstrates that optimizing these properties leads to improved performance in downstream tasks and provides a theoretical framework for understanding contrastive learning's effectiveness in representation learning. Together, these works lay the groundwork for a deeper theoretical understanding of contrastive learning, highlighting the necessity for additional investigation.

## 3 PRELIMINARIES

### 3.1 METRIC SPACE

A metric space is a set $X$ together with a distance function $d : X \times X \to \mathbb{R}$ (called a metric) that satisfies the following properties for all $x, y, z \in X$:

1. Non-negativity: $d(x, y) \geq 0$
2. Identity of indiscernibles: $d(x, y) = 0$ if and only if $x = y$
3. Symmetry: $d(x, y) = d(y, x)$
4. Triangle inequality: $d(x, z) \leq d(x, y) + d(y, z)$

### 3.2 SEMI-METRIC SPACE

A semi-metric space is a generalization of a metric space where the triangle inequality is not required to hold. It is defined as a set $X$ with a distance function $d : X \times X \to \mathbb{R}$ that satisfies:

1. Non-negativity: $d(x, y) \geq 0$
2. Identity of indiscernibles: $d(x, y) = 0$ if and only if $x = y$
3. Symmetry: $d(x, y) = d(y, x)$

## 4 METHOD

### 4.1 SIMILARITY-ORTHOGONALITY (SIMO) LOSS FUNCTION

We propose a novel loss function that leverages Euclidean distance and orthogonality (through the squared dot product) for learning the embedding space. This function, which we term the Similarity-Orthogonality (SimO) loss, is defined as:

$$\mathcal{L}_{\text{SimO}} = y \left[ \frac{\sum_{i,j} d_{ij}}{\epsilon + \sum_{i,j} o_{ij}} \right] + (1 - y) \left[ \frac{\sum_{i,j} o_{ij}}{\epsilon + \sum_{i,j} d_{ij}} \right] \tag{1}$$

- $\forall i, j, \ i \neq j \ and \ i \leq j$ are indices of the embedding pairs within a batch

- $y$ is a binary label for the entire batch, where $y = 1$ for similarity and $y = 0$ for dissimilarity

- $d_{ij} = ||e_i - e_j||_2^2 \cdot e_j$ is the squared Euclidean distance between embeddings $e_i$ and $e_j$

- $o_{ij} = (e_i \cdot e_j)^2$ is the squared dot product of embeddings $e_i$ and $e_j$

- $\epsilon$ is a small constant to prevent division by zero

SimO loss function presents a novel framework for learning embedding spaces, addressing several critical challenges in representation learning. Below, we highlight its key properties and advantages:

- Semi-Metric Space function: The SimO loss function operates within a semi-metric space, as formalized in the SimO Semi-Metric Space Theorem. This allows for a flexible representation of distances between embeddings, particularly useful for high-dimensional data where traditional metrics may fail to capture complex relationships (Theorem **??**).

- Preventing Dimensionality Collapse: The orthogonality component of the SimO loss plays a pivotal role in preventing dimensionality collapse, a phenomenon where dissimilar classes become indistinguishable in the embedding space. By encouraging orthogonal embeddings for distinct classes, SimO ensures that the learned representations remain well-separated and span diverse regions of the embedding space, preserving class distinctiveness (Theorem A.2).

---

**Algorithm 1** SIMO Loss Function

---

1: **Input:** embeddings, label_batch, indices, epsilon
2: **function** orthogonality_loss(embeddings, indices)
   \# indices contains the unique combinations between different embeddings
3: $E1 \leftarrow$ embeddings[indices[0]]
4: $E2 \leftarrow$ embeddings[indices[1]]
5: dot_product_squared $\leftarrow$ vmap(pairwise_dot_product_squared)
6: loss $\leftarrow \sum$ dot_product_squared($E1, E2$)
7: **return** loss
8: **function** similarity_loss(embeddings, indices)
   \# indices contains the unique combinations between different embeddings
9: $E1 \leftarrow$ embeddings[indices[0]]
10: $E2 \leftarrow$ embeddings[indices[1]]
11: squared_distance $\leftarrow$ vmap(pairwise_squared_distance)
12: loss $\leftarrow \sum$ squared_distance($e1, e2$)
13: **return** loss
14: **function** SimO_Loss(embeddings, label_batch, indices, epsilon)
15: ortho_loss $\leftarrow$ orthogonality_loss(embeddings, indices)
16: sim_loss $\leftarrow$ similarity_loss(embeddings, indices)
17: total_loss $\leftarrow$ label_batch $\cdot \frac{\text{sim\_loss}}{\epsilon + \text{ortho\_loss}} + (1 - \text{label\_batch}) \cdot \frac{\text{ortho\_loss}}{\epsilon + \text{sim\_loss}}$
18: **return** total_loss/indices[0].shape[0]

---

- Mitigating the Curse of Orthogonality: Our embedding techniques are constrained by the Curse of Orthogonality, which limits the number of mutually orthogonal vectors to the dimensionality of the embedding space. SimO overcomes this limitation by leveraging orthogonality-based regularization (orthogonality leaning factor) informed by the Johnson-Lindenstrauss lemma (Theorem A.4), thus enabling more effective utilization of the available space without falling prey to orthogonality saturation (Theorem A.3).

### 4.2 ANCHOR-FREE CONTRASTIVE LEARNING

We introduce a novel contrastive learning pretraining strategy (Algorithm 2) that uses the SimO loss function. For each iteration, we create a batch of $k$ images sampled from n randomly selected classes where $num\_classes = batch\_size//k$. We generate the embeddings using our model.

The loss computation strategy is the sum of three different operations:

- To calculate the loss over embeddings from the same class, we reshape the embeddings to ($num\_classes, k, embeddings\_dim$):

$\mathcal{L}_{\text{same}} = \text{SimO\_loss}(embeddings, 1.0)$ SimO is applied class-wise (axis 0) to calculate the loss over similar embeddings then sum it up.

- Using the same *embeddings* predicted, we continue to do the following:

  - Compute the mean embedding for each of the $n_{classes}$ represented in the batch: $\mu_i = \frac{1}{k} \sum_{j=1}^{k} f_\theta(I_j)$, where $I_j \in mb_i$
  - Calculate the loss using these mean embeddings: $\mathcal{L}_{mean\_dissimilar} = \text{simo}([\mu_1, \mu_2, ..., \mu_m], 0 + olean)$, with *olean* is the orthogonality leaning factor

- For $\mathcal{L}_{dissimilar}$ We reshape the embeddings to ($k, num\_classes, embeddings\_dim$) and then we calculate $\mathcal{L}_{dissimilar} = \text{SimO\_loss}(embeddings, 0.0 + olean)$ where *olean* is the orthogonality leaning factor.

This dual-batch approach allows the model to learn both intra-class compactness (through $\mathcal{L}_{\text{same}}$) and inter-class separability (through $\mathcal{L}_{mean\_dissimilar}$ and $\mathcal{L}_{dissimilar}$). By oper-

---

**Algorithm 2** Anchor-Free Contrastive Learning with SimO loss pseudo-implementation

---

1: **Input:** data $(x_{train}, y_{train})$, num_epochs, batch_size, num_classes, k, olean
2: Initialize model $f$ parameters $\theta$
3: Initialize optimizer state
4: **for** $iteration = 1$ **to** num_iterations **do**
5:     batch, labels $\leftarrow$ create_mean_batch(data, batch_size, k, num_classes)
6:     $embeddings \leftarrow f_\theta(batch; \theta)$
7:     $embeddings \leftarrow embeddings.reshape(num\_classes, k, embeddings\_dim)$ # Reshaping the embeddings to group similar images together
8:     $\mathcal{L}\_similar \leftarrow simo\_loss(embeddings, 1.0)$ # No need for orthogonality leaning
9:     $mean\_embeddings \leftarrow \{\}$
10:     **for** each unique label in $labels$ **do**
11:       $mean\_e \leftarrow mean(embeddings[\text{labels} == label])$
12:       $mean\_embedding \leftarrow mean\_embeddings \cup \{mean\_e\}$
13:     **end for**
14:     $\mathcal{L}\_mean\_dissimilar \leftarrow simo\_loss(mean\_embeddings, 0 + olean)$
15:     $embeddings \leftarrow embeddings.reshape(k, num\_classes, embeddings\_dim)$ # Reshaping the Embeddings to group dissimilar images together
16:     $\mathcal{L}\_dissimilar \leftarrow simo\_loss(embeddings, 0.0 + olean)$ # $olean$ is Orthogonality leaning
17:     $\mathcal{L} \leftarrow \mathcal{L}\_similar + \mathcal{L}\_mean\_dissimilar + \mathcal{L}\_dissimilar$
18:     $g \leftarrow \nabla_\theta \mathcal{L}$
19:     Update $\theta$ and optimizer state using $g$
20: **end for**
21: **return** Trained model parameters $\theta$

---

ating on class means, we encourage the model to learn more robust discriminative features that generalize well across class instances with reduce the impact of negative sampling. The overall training objective alternates between these two batch types, optimizing: $\mathcal{L} = \mathbb{E}[\mathcal{L}_{same}] + \mathbb{E}[\mathcal{L}_{mean\_different}] + \mathbb{E}[\mathcal{L}_{dissimilar}]$ where the expectation is taken over the random sampling of batches during training.

## 4.3 EXPERIMENTAL SETUP

Our experiments were conducted using GPU-enabled cloud computing platforms, with experiment tracking and visualization handled by Weights & Biases Biewald (2020). We implemented our models using JAX/Flax and TensorFlow frameworks. For our experiments, we utilized the CIFAR-10 dataset Krizhevsky & Hinton (2009). CIFAR-10 consists of 60,000 32x32 RGB images across 10 classes, with 50,000 for training and 10,000 for testing.

In the pretraining phase for CIFAR-10, we used an embedding dimension of 16 for the linear projection head following the ResNet encoder with layer normalization instead of batch normalization, with a batch size of 96 and 32 randomly selected images per class from 3 classes. During the linear probing phase, we fed the projection of our frozen pretrained model to a classifier head consisting of one MLP layer with 128 neurons, followed by an output layer matching the number of classes in CIFAR-10 dataset.

## 5 RESULTS

Our extensive experiments on the CIFAR-10 dataset demonstrate the effectiveness of SimO in learning discriminative and interpretable embeddings. We present a multifaceted analysis of our results, encompassing unsupervised clustering, supervised fine-tuning, and qualitative visualization of the embedding space.

To assess the transferability and discriminative capacity of our learned representations, we conducted a supervised fine-tuning experiment. We froze the SimO-trained encoder and attached a simple classifier head, fine-tuning only this newly added layer for a single epoch. This minimal fine-tuning yielded impressive results:

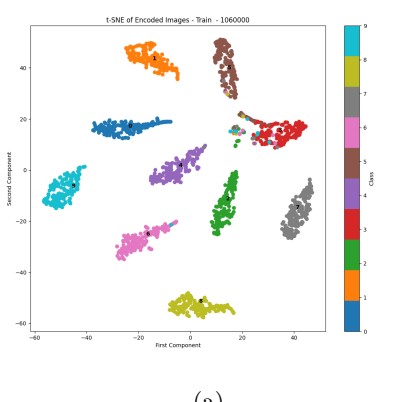 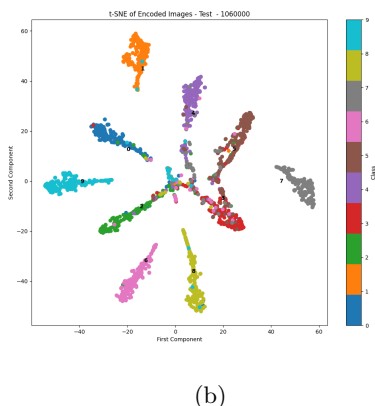

(a)        (b)

Figure 2: Manifold visualization of the Embedding Space using T-SNE for both (a) trainset and (b) testset

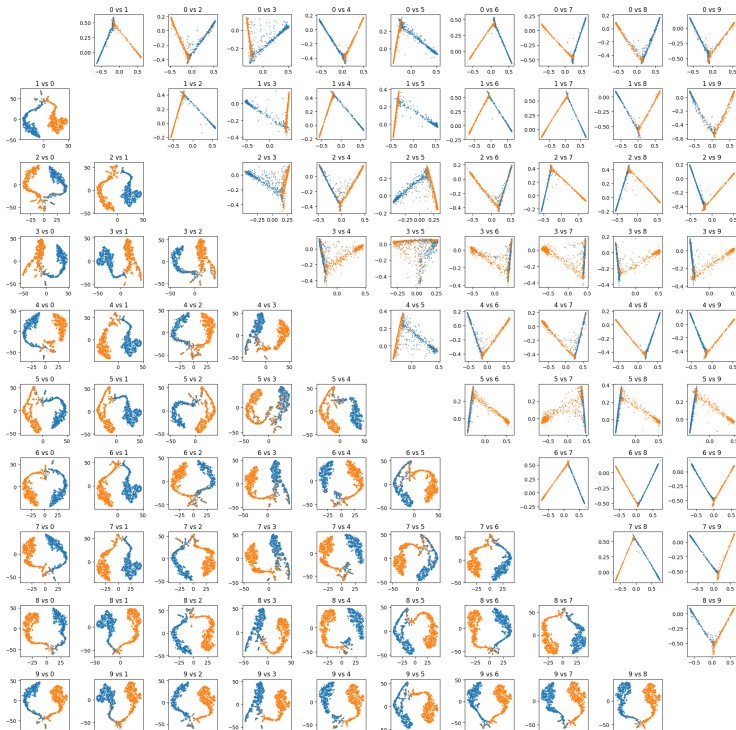

Figure 3: Pairwise Manifold Visualization using TSNE (Lower-Triangular Plots and PCA (Uper Triangular Plots)

The rapid convergence to high accuracy with minimal fine-tuning underscores the quality and transferability of our SimO-learned representations. It's worth noting that this performance was achieved with only 1 epoch of fine-tuning, demonstrating the efficiency of our approach.

Examination of the confusion matrix revealed that the model primarily struggles with distinguishing between the 'cat' and 'dog' classes. This observation aligns with our qualitative analysis of the embedding space visualizations (Figure 2, Figure 3, Figure 4). The challenge in separating these classes is not unexpected, given the visual similarities between cats and dogs, and has been observed in previous works Khosla et al. (2012); Zhang et al. (2021).

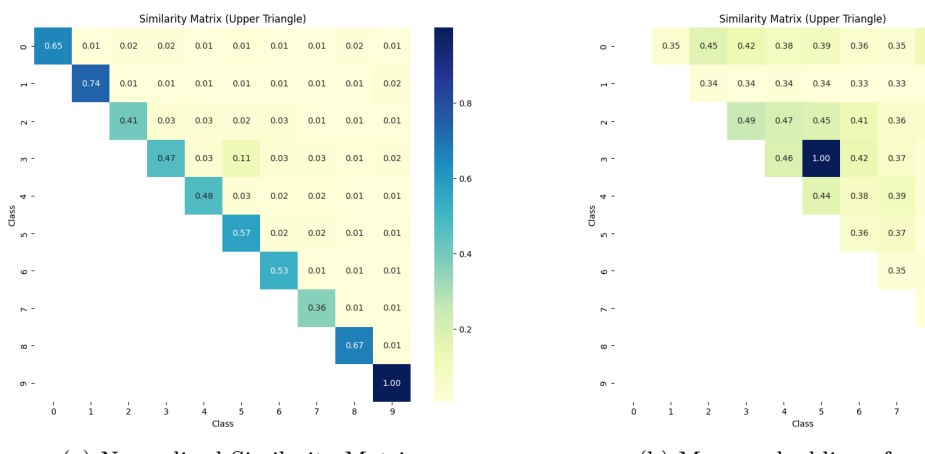

(a) Normalized Similarity Matrix    (b) Mean embedding of each class

Figure 4: Normalized Similarity Matrix calculated using SimO (a) Pairwise embeddings (b) Class Means

| Dataset | Model | Projection Head dim. | Train Accuracy (%) | Test Accuracy (%) |
|---------|-------|----------------------|--------------------|--------------------|
| Cifar10 | ResNet18 | 16 | 94 | 85 |

Table 1: Model Performance Metrics over 1 epoch fine-tuning of a classifier head and the frozen pretrained model

We conducted a longitudinal analysis of the embedding space evolution using t-SNE projections at various training iterations (Figure 5). This analysis revealed intriguing dynamics in the learning process:

**Continual Learning Behavior:** We observed a tendency towards continual learning (Figure 5), where the model appeared to focus on one class at a time. This behavior suggests that SimO naturally induces a curriculum-like learning process, potentially contributing to its effectiveness.

**Persistent Challenges:** The 'cat' and 'dog' classes remained challenging for the model from 100,000 iterations up to 1 million iterations. This persistent difficulty aligns with our quantitative error analysis and highlights an area for potential future improvements.

**Progressive Separation:** For training, we observed a clear trend of increasing inter-class separation and intra-class cohesion, with the exception of the aforementioned challenging classes.

These results collectively demonstrate the efficacy of SimO in learning rich, discriminative, and interpretable embeddings. The observed continual learning behavior and the challenges with visually similar classes provide insights into the learning dynamics of our approach and point to exciting directions for future research.

## 6 ABLATION STUDY

Our ablation studies provide crucial insights into the effectiveness of SimO's key components.

**Orthogonality Leaning Factor** When removing the orthogonality constraint from our loss function, we observed a significant degradation in performance. The model failed to learn representations for all classes, instead converging to a state where only 4 out of 10 classes from CIFAR-10 were distinguishable, with the remaining 6 classes grouped together. This result underscores the critical role of the orthogonality leaning factor in SimO, acting as a regularizer that encourages the model to utilize the full dimensionality of the embedding space and prevent the clustering of multiple classes into a single region which validate.

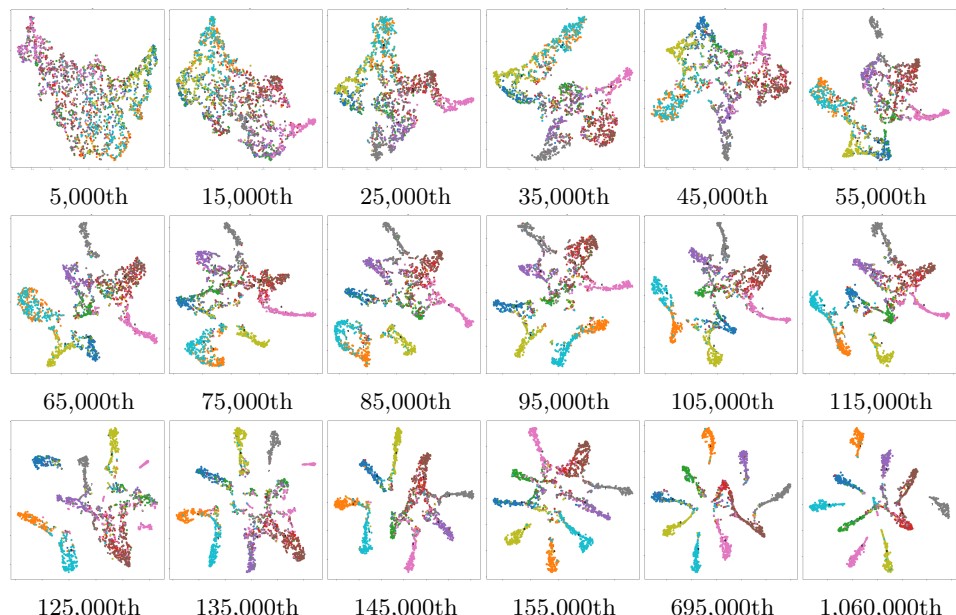

Figure 5: Continual Learning Properties of SimO with AFCL Framework

Number of Classes per Batch: In our initial experiments with batch composition, we encountered an interesting phenomenon where certain classes dominated the loss function and, consequently, the embedding space. This dominance prevented the model from learning adequate representations for the other classes, resulting in unstable learning. To address this issue, we implemented a class sampling strategy inspired by techniques used in meta-learning, randomly selecting less than 50% of the total number of classes for each batch. This approach led to more balanced learning across all classes increasing stability in the learning process.

**Lower Bound of Embedding Dimension** To explore the lower bounds of the embedding dimension and understand the compressive capabilities of SimO, we conducted an experiment where we pretrained a ResNet18 model with a projection head outputting only 2 dimensions, maintaining the orthogonality learning component. Remarkably, we achieved a clustering accuracy of 60% on CIFAR-10 with this extreme dimensionality reduction. This result is particularly impressive given that standard contrastive learning methods typically struggle with such low dimensions, often failing to separate classes meaningfully. This demonstrates the power of SimO's orthogonality constraint in creating discriminative embeddings even in very low-dimensional spaces, pointing to its potential in scenarios where compact representations are required.

## 7 DISCUSSION AND LIMITATIONS

In our proposed framework, the embedding space generated by the SimO loss exhibits notable geometric properties (Figure 2, Figure 3) that can be interpreted through the lenses of stratified spaces, quotient topology, and fiber bundles. Specifically, we can view the overall embedding space as a stratified space, where each stratum corresponds to a distinct class neighborhood. This structure is facilitated by the orthogonality encouraged by our loss function, promoting clear separations between classes while maintaining cohesive intra-class relationships. Furthermore, we propose considering a quotient topology in which points within the same class neighborhood are identified, simplifying the representation of the embedding space to a point for each class. This transformation not only highlights the distinctness of classes but also emphasizes their orthogonality in the learned space. Additionally, our method generates a structure reminiscent of a fiber bundle, where each fiber corresponds to a specific class and is orthogonal to other fibers. This fiber bundle-like

organization allows for a rich representation of class relationships and facilitates a more interpretable understanding of the learned embeddings. Collectively, these geometric interpretations underscore the robustness and effectiveness of our SimO loss with our AFCL framework in structuring embeddings that balance class separation with interpretability requiring small batch sizes unlike other loss functions.

While SimO demonstrates significant advancements in contrastive learning, our extensive experimentation has revealed several important limitations and areas for future research.

**Redefinition of Similarity Metrics**: A key finding of our work is that embeddings learned through SimO no longer adhere to traditional similarity measures such as cosine similarity. This departure from conventional metrics necessitates a paradigm shift in how we evaluate similarity in the embedding space. Our SimO loss itself emerges as the most appropriate measure of similarity or dissimilarity between embeddings. This also presents challenges for integration with existing systems and methods that rely on cosine similarity. Future work should focus on developing efficient computational methods for this new similarity metric and exploring its theoretical properties.

**Sensitivity to Data Biases**: Our method's ability to learn fine-grained representations comes with increased sensitivity to biases present in the training data. This is particularly evident in the case of background biases in object recognition tasks. For instance, our model struggled to separate the neighborhoods of Dog and Cat classes even though it learned from the 120,000th iteration to the 1 millionth iteration, despite having learned most other classes effectively. This sensitivity necessitates robust data augmentation techniques to mitigate the impact of such biases. While this requirement for strong augmentation can be seen as a limitation, it also highlights SimO's potential for detecting and quantifying dataset biases, which could be valuable for improving dataset quality and fairness in machine learning models.

**The Orthogonality Learning Factor**: The performance of SimO is notably influenced by the orthogonality learning factor, a hyperparameter that balances the trade-off between similarity and orthogonality objectives. Finding the optimal value for this factor presents a challenge we term "the curse of orthogonality." Too low a factor leads to insufficient separation between class neighborhoods, while too high a factor can result in overly rigid embeddings that fail to capture intra-class variations. Our experiments show that this factor often needs to be tuned specifically for each dataset and task, which can be computationally expensive. Developing adaptive methods for automatically adjusting this factor during training represents an important direction for future research.

**Computational Complexity**: While not unique to SimO, the computational requirements for optimizing orthogonality in high-dimensional spaces are substantial. This can limit the applicability of our method to very large datasets or in resource-constrained environments. Future work should explore approximation techniques or more efficient optimization strategies to address this limitation.

Despite these limitations, we believe that SimO represents a significant step forward in contrastive learning. The challenges identified here open up exciting new avenues for research in representation learning, similarity metrics, and bias mitigation in machine learning models. Addressing these limitations will not only improve SimO but also deepen our understanding of the fundamental principles underlying effective representation learning.

## 8 Conclusion

Our AFCL method introduces the SimO loss function as a novel approach to contrastive learning, effectively addressing several critical challenges related to embedding space utilization and interoperability. By optimizing both the similarity and orthogonality of embeddings, SimO prevents dimensionality collapse and ensures that class representations remain distinct, even in lower dimensions, requiring smaller batch sizes and embedding dimensions.

Our experimental results on the CIFAR-10 dataset demonstrate the efficacy of SimO in generating structured and discriminative embeddings with minimal computational over-

head. Notably, our method achieves impressive test accuracy as early as the first epoch. Although there are limitations, such as sensitivity to data biases and dependence on specific hyperparameters, SimO paves the way for future advancements in enhancing contrastive learning techniques and managing embedding spaces more effectively.

## REPRODUCIBILITY STATEMENT

We provide detailed proof for all the lemmas and theorems in the Appendices. Code will be shared publicly after publication.

## LICENSE

The source code, algorithms, and all contributions presented in this work are licensed under the GNU Affero General Public License (AGPL) v3.0. This license ensures that any use, modification, or distribution of the code and any adaptations or applications of the underlying models and methods must be made publicly available under the same license. This applies whether the work is used for personal, academic, or commercial purposes, including services provided over a network.

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

## A  APPENDIX

**Theorem A.1** ($\overline{\mathbb{E}}$ is metric and $\mathbb{E}$ is pseudo-metric). *Let $(\mathbb{R}^n, \overline{\mathbb{E}})$ and $(\mathbb{R}^n, \mathbb{E})$ be two spaces where:*

$$\mathbb{E}(e_i, e_j) = \frac{o_{ij}}{d_{ij}} = \frac{(e_i \cdot e_j)^2}{||e_i - e_j||^2}$$

$$\overline{\mathbb{E}}(e_i, e_j) = \frac{d_{ij}}{o_{ij}} = \frac{||e_i - e_j||^2}{(e_i \cdot e_j)^2}$$

*for $e_i, e_j \in \mathbb{R}^n \setminus \{0\}$ where $e_i \neq e_j$, with $d_{ij} = ||e_i - e_j||^2$ being the squared Euclidean distance and $o_{ij} = (e_i \cdot e_j)^2$ being the squared dot product.*

*Then $(\mathbb{R}^n, \overline{\mathbb{E}})$ is pseudo-metric when $e_i \not\perp e_j$, while $(\mathbb{R}^n, \mathbb{E})$ is pseudo-metric.*

*Proof.* We structure this proof into four parts:

1. Preliminary observations and domain analysis

2. Proof of common properties for both measures

3. Proof that $\mathbb{E}$ (yat) is a pseudo-metric

4. Proof that $\overline{\mathbb{E}}$ (posi-yat) is a metric

**Part I: Preliminary Observations**

Before proving the metric properties, we must establish the domain where these measures are well-defined:

1. For non-zero vectors $e_i, e_j$:

- $d_{ij}^2 = 0 \iff e_i = e_j$

- $o_{ij} = 0 \iff e_i \perp e_j$ (vectors are orthogonal)

2. Domain restrictions:

- $\mathbb{E}$ is defined when $d_{ij}^2 \neq 0$ (distinct vectors)

- $\overline{\mathbb{E}}$ is defined when $o_{ij} \neq 0$ (non-orthogonal vectors)

**Part II: Common Properties**

Both measures satisfy the following properties:

**1. Non-negativity:** Since both $d_{ij}^2$ and $o_{ij}$ are squared quantities:

$$d_{ij}^2 = ||e_i - e_j||^2 \geq 0 \quad \text{and} \quad o_{ij} = (e_i \cdot e_j)^2 \geq 0$$

Therefore:

$$\mathbb{E}(e_i, e_j) \geq 0 \quad \text{and} \quad \overline{\mathbb{E}}(e_i, e_j) \geq 0$$

**2. Identity of Indiscernibles:** For both measures, we prove this bidirectionally:

($\Rightarrow$) If $e_i = e_j$:

- $d_{ij}^2 = 0$

- $o_{ij} = ||e_i||^4 > 0$ (for non-zero vectors)

Therefore, $\mathbb{E}(e_i, e_j) = 0$ and $\overline{\mathbb{E}}(e_i, e_j) = 0$

($\Leftarrow$) If $\mathbb{E}(e_i, e_j) = 0$ or $\overline{\mathbb{E}}(e_i, e_j) = 0$:

- For $\mathbb{E}$: $\frac{o_{ij}}{d_{ij}^2} = 0 \implies o_{ij} = 0$ (since $d_{ij}^2 \neq 0$ for distinct vectors)

- For $\overline{\mathbb{E}}$: $\frac{d_{ij}^2}{o_{ij}} = 0 \implies d_{ij}^2 = 0$ (since $o_{ij} \neq 0$ in domain)

In both cases, this implies that this rule stands for $\overline{\mathbb{E}}$, but not for $\mathbb{E}$.

**3. Symmetry:** Symmetry follows from the symmetry of the dot product and Euclidean distance:

$$\mathbb{E}(e_i, e_j) = \frac{(e_i \cdot e_j)^2}{||e_i - e_j||^2} = \frac{(e_j \cdot e_i)^2}{||e_j - e_i||^2} = \mathbb{E}(e_j, e_i)$$

And similarly for $\overline{\mathbb{E}}$.

**Part III: Proof the triangle inequality for $\mathbb{E}$**

To prove $\mathbb{E}$ satisfies the triangle inequality, we proceed in steps:

Given:

$$\mathbf{e}_1 \mathbb{E} \mathbf{e}_2 = \frac{(\mathbf{e}_1 \cdot \mathbf{e}_2)^2}{||\mathbf{e}_2 - \mathbf{e}_1||^2}.$$

Let:

- $\mathbf{e}_1$ and $\mathbf{e}_2$ be vectors in $\mathbb{R}^n$,

- $\theta$ be the angle between $\mathbf{e}_1$ and $\mathbf{e}_2$.

The dot product between $\mathbf{e}_1$ and $\mathbf{e}_2$ can be written as:

$$\mathbf{e}_1 \cdot \mathbf{e}_2 = ||\mathbf{e}_1|| \, ||\mathbf{e}_2|| \cos \theta.$$

Thus, $(\mathbf{e}_1 \cdot \mathbf{e}_2)^2$ becomes:

$$(\mathbf{e}_1 \cdot \mathbf{e}_2)^2 = (||\mathbf{e}_1|| \, ||\mathbf{e}_2|| \cos \theta)^2 = ||\mathbf{e}_1||^2 \, ||\mathbf{e}_2||^2 \cos^2 \theta.$$

The Euclidean distance between $\mathbf{e}_1$ and $\mathbf{e}_2$ is:

$$||\mathbf{e}_2 - \mathbf{e}_1||^2 = ||\mathbf{e}_1||^2 + ||\mathbf{e}_2||^2 - 2 ||\mathbf{e}_1|| \, ||\mathbf{e}_2|| \cos \theta.$$

Now we substitute these expressions into the formula for $\mathbf{e}_1 \boxminus \mathbf{e}_2$:

$$\mathbf{e}_1 \boxminus \mathbf{e}_2 = \frac{||\mathbf{e}_1||^2 \, ||\mathbf{e}_2||^2 \cos^2 \theta}{||\mathbf{e}_1||^2 + ||\mathbf{e}_2||^2 - 2 ||\mathbf{e}_1|| \, ||\mathbf{e}_2|| \cos \theta}.$$

Let's simplify by defining:

- $A = ||\mathbf{e}_1||$,

- $B = ||\mathbf{e}_2||$.

Thus, the expression becomes:

$$f(\theta) = \mathbf{e}_1 \boxminus \mathbf{e}_2 = \frac{A^2 B^2 \cos^2 \theta}{A^2 + B^2 - 2AB \cos \theta}.$$

Let's factor out common terms in the numerator. Notice that each term in the numerator has a factor of $A^2 B^2 \sin \theta$, so we can factor that out:

$$f'(\theta) = \frac{A^2 B^2 \sin \theta \left[ -2 \cos \theta (A^2 + B^2 - 2AB \cos \theta) - 2AB \cos^2 \theta \right]}{(A^2 + B^2 - 2AB \cos \theta)^2}.$$

Now, distribute $-2 \cos \theta$ in the first term inside the brackets:

$$= \frac{A^2 B^2 \sin \theta \left[ -2A^2 \cos \theta - 2B^2 \cos \theta + 4AB \cos^2 \theta - 2AB \cos^2 \theta \right]}{(A^2 + B^2 - 2AB \cos \theta)^2}.$$

Combine the $\cos^2 \theta$ terms:

$$= \frac{A^2 B^2 \sin \theta \left[ -2A^2 \cos \theta - 2B^2 \cos \theta + 2AB \cos^2 \theta \right]}{(A^2 + B^2 - 2AB \cos \theta)^2}.$$

Thus, the simplified form of $f'(\theta)$ is:

$$f'(\theta) = \frac{-2A^2 B^2 \sin \theta \left( A^2 \cos \theta + B^2 \cos \theta - AB \cos^2 \theta \right)}{(A^2 + B^2 - 2AB \cos \theta)^2}.$$

This form is simpler and allows us to see that the sign of $f'(\theta)$ depends on the sign of $-\sin \theta$, which is non-positive on the interval $[0, \pi]$. Therefore, $f'(\theta) \leq 0$ on this interval, confirming that $f(\theta)$ is monotonically decreasing.

Since $f(\theta)$ is monotonically decreasing, it follows that $\mathbb{E}(\mathbf{e}_1, \mathbf{e}_2) = f(\theta)$ decreases as $\theta$ increases.

Applying the Angular Triangle Inequality Angles in Euclidean space satisfy the triangle inequality (Cauchy–Schwarz inequality):

$$\theta_{ik} \leq \theta_{ij} + \theta_{jk}.$$

Since $\mathbb{E}(\mathbf{e}_i, \mathbf{e}_j)$ is a decreasing function of $\theta$, we conclude:

$$\mathbb{E}(\mathbf{e}_i, \mathbf{e}_k) \leq \mathbb{E}(\mathbf{e}_i, \mathbf{e}_j) + \mathbb{E}(\mathbf{e}_j, \mathbf{e}_k).$$

Since $\mathbb{E}(\mathbf{e}_i, \mathbf{e}_j) = \frac{1}{\overline{\mathbb{E}}(\mathbf{e}_i, \mathbf{e}_j)}$, we deduce that the $\overline{\mathbb{E}}$ is a function of $\theta$, we conclude:

$$\overline{\mathbb{E}}(\mathbf{e}_i, \mathbf{e}_k) \leq \overline{\mathbb{E}}(\mathbf{e}_i, \mathbf{e}_j) + \overline{\mathbb{E}}(\mathbf{e}_j, \mathbf{e}_k).$$

, the only problem that is preventing the $\overline{\mathbb{E}}$ from being a fully metric space is not being defined when $e_i \perp e_j$, but we can remedy this with an $\epsilon$ so now it becomes $\overline{\mathbb{E}}' = \frac{d_{ij}^2}{\epsilon + o_{ij}}$, we do the same thing to the $\mathbb{E}$ to define it when $e_i = e_j$, just unlike the $\overline{\mathbb{E}}$, this operation doesn't change the fact that $\mathbb{E}$ remains pseudo-metric.

$\square$

**Theorem A.2** (SimO Dimentionality Collapse Prevention). *The loss function $\mathcal{L}_{SimO}$ prevents dimensionality collapse for dissimilar (negative) classes through its orthogonality term.*

*Proof.* Let $\mathcal{E} = \{e_1, \ldots, e_n\}$ be a set of embeddings in $\mathbb{R}^d$, where $n$ is the batch size and $d$ is the embedding dimension.

The loss function $\mathcal{L}_{\text{SimO}}$ is defined as:

$$\mathcal{L}_{\text{SimO}} = y \left[ \frac{\sum_{i,j} d_{ij}}{\epsilon + \sum_{i,j} o_{ij}} \right] + (1 - y) \left[ \frac{\sum_{i,j} o_{ij}}{\epsilon + \sum_{i,j} d_{ij}} \right] \tag{2}$$

where:

- $i, j \in \{1, \ldots, n\}$, $i \neq j$, $i < j$

- $y \in \{0, 1\}$ is a binary label for the entire batch (1 for similarity, 0 for dissimilarity)

- $d_{ij} = \|e_i - e_j\|^2 = \|e_i\|^2 + \|e_j\|^2 - 2e_i \cdot e_j$ is the squared Euclidean distance

- $o_{ij} = (e_i \cdot e_j)^2$ is the squared dot product

- $\epsilon > 0$ is a small constant to prevent division by zero

We proceed by analyzing the behavior of $\mathcal{L}_{\text{SimO}}$ for dissimilar pairs and showing how it encourages properties that prevent dimensionality collapse.

1. For dissimilar pairs ($y = 0$), $\mathcal{L}_{\text{SimO}}$ reduces to:

$$\mathcal{L}_{\text{SimO}} = \frac{\sum_{i,j} o_{ij}}{\epsilon + \sum_{i,j} d_{ij}} \tag{3}$$

2. To minimize this loss, we must minimize $\sum_{i,j} o_{ij}$ and maximize $\sum_{i,j} d_{ij}$.

3. We first prove two key lemmas:

   **Lemma A.2.1.** *Minimizing $\sum_{i,j} o_{ij}$ encourages orthogonality between dissimilar embeddings.*

*Proof.*  • $\forall i, j : o_{ij} = (e_i \cdot e_j)^2 \geq 0$
  • Minimizing $\sum_{i,j} o_{ij}$ implies minimizing each $o_{ij}$
  • Minimizing $(e_i \cdot e_j)^2$ pushes $e_i \cdot e_j \to 0$
  • $e_i \cdot e_j = 0 \iff e_i \perp e_j$

Therefore, minimizing $\sum_{i,j} o_{ij}$ encourages orthogonality between all pairs of dissimilar embeddings. $\qquad\square$

**Lemma A.2.2.** *Maximizing $\sum_{i,j} d_{ij}$ encourages dissimilar embeddings to be far apart in the embedding space.*

*Proof.*  • $\forall i, j : d_{ij} = \|e_i - e_j\|^2 \geq 0$
  • Maximizing $\sum_{i,j} d_{ij}$ implies maximizing each $d_{ij}$
  • Maximizing $\|e_i - e_j\|^2$ increases the Euclidean distance between $e_i$ and $e_j$

Therefore, maximizing $\sum_{i,j} d_{ij}$ pushes dissimilar embeddings farther apart in the embedding space. $\qquad\square$

4. Now, we show how these lemmas prevent dimensionality collapse:

   (a) By Lemma 1, $\mathcal{L}_{\mathrm{SimO}}$ encourages orthogonality between dissimilar embeddings.
       • Orthogonal vectors span different dimensions in the embedding space.
       • This prevents dissimilar embeddings from aligning along the same dimensions.
   (b) By Lemma 2, $\mathcal{L}_{\mathrm{SimO}}$ simultaneously pushes dissimilar embeddings farther apart.
       • This reinforces the distinctiveness of dissimilar embeddings.
       • It prevents dissimilar embeddings from collapsing to nearby points in the embedding space.
   (c) The combination of (a) and (b) ensures that:
       • Dissimilar embeddings maintain their distinctiveness.
       • They occupy different regions and directions in the embedding space.
       • The effective dimensionality of the embedding space is preserved for dissimilar classes.

5. Formally, let $\{e_i\}_{i=1}^k$ be a set of dissimilar embeddings. The loss function ensures:

   • $\forall i \neq j : e_i \cdot e_j \approx 0$ (orthogonality)
   • $\forall i \neq j : \|e_i - e_j\|^2$ is maximized (separation)

   These conditions directly contradict the definition of dimensionality collapse, where dissimilar embeddings would become very similar or identical.

$\qquad\square$

**Theorem A.3** (Curse of Orthogonality). *In an $n$-dimensional embedding space, the number of classes that can be represented with mutually orthogonal embeddings is at most $n$.*

*Proof.* Let $\mathbb{R}^n$ be an $n$-dimensional embedding space. Consider a set of $k$ vectors $\{\mathbf{a}_1, \mathbf{a}_2, \ldots, \mathbf{a}_k\}$ in this space, where each vector represents the mean embedding of a distinct class.

Assume that these vectors are mutually orthogonal, i.e.,

$$\mathbf{a}_i \cdot \mathbf{a}_j = 0 \quad \text{for all} \quad 1 \leq i, j \leq k \quad \text{and} \quad i \neq j. \tag{4}$$

In $\mathbb{R}^n$, the maximum number of mutually orthogonal vectors is equal to the dimension of the space, which is $n$. This is because any set of mutually orthogonal vectors must also be linearly independent, and the maximum number of linearly independent vectors in $\mathbb{R}^n$ is $n$.

Therefore, the maximum number of mutually orthogonal embeddings, and thus the maximum number of classes that can be represented with such embeddings, is $n$.

Hence, $k \leq n$.

$\square$

### A.1 Johnson–Lindenstrauss Lemma

**Lemma A.3.1** (Johnson–Lindenstrauss Lemma). *$0 < \epsilon < 1$, and let $X$ be a set of $n$ points in $\mathbb{R}^d$. There exists a mapping $f : \mathbb{R}^d \to \mathbb{R}^k$ with $k = O\left(\frac{\log n}{\epsilon^2}\right)$ such that for all $x, y \in X$:*

$$(1 - \epsilon)\|x - y\|^2 \leq \|f(x) - f(y)\|^2 \leq (1 + \epsilon)\|x - y\|^2$$

**Theorem A.4** (Johnson-Lindenstrauss Lemma Addressing the Curse of Orthogonality). *Given $k > n$ vectors in $\mathbb{R}^n$, there exists a projection into a higher-dimensional space $\mathbb{R}^m$ where $m = O(\frac{\log k}{\epsilon^2})$, such that the projected vectors are "nearly orthogonal", effectively overcoming the limitation imposed by the Curse of Orthogonality.*

*Proof.* Let $\{v_1, v_2, \ldots, v_k\}$ be $k$ vectors in $\mathbb{R}^n$, where $k > n$.

1) By the Johnson-Lindenstrauss lemma, there exists a mapping $f : \mathbb{R}^n \to \mathbb{R}^m$, where $m = O(\frac{\log k}{\epsilon^2})$, such that for all $i, j \in \{1, \ldots, k\}$:

$$(1 - \epsilon)\|v_i - v_j\|^2 \leq \|f(v_i) - f(v_j)\|^2 \leq (1 + \epsilon)\|v_i - v_j\|^2$$

2) Consider the dot product of two projected vectors $f(v_i)$ and $f(v_j)$ for $i \neq j$:

$$f(v_i) \cdot f(v_j) = \frac{1}{2}(\|f(v_i)\|^2 + \|f(v_j)\|^2 - \|f(v_i) - f(v_j)\|^2)$$

3) Using the upper bound from the JL lemma:

$$f(v_i) \cdot f(v_j) \leq \frac{1}{2}(\|f(v_i)\|^2 + \|f(v_j)\|^2 - (1 - \epsilon)\|v_i - v_j\|^2)$$

4) If $v_i$ and $v_j$ were originally orthogonal, then $\|v_i - v_j\|^2 = \|v_i\|^2 + \|v_j\|^2$. Substituting this:

$$f(v_i) \cdot f(v_j) \leq \frac{1}{2}(\|f(v_i)\|^2 + \|f(v_j)\|^2 - (1 - \epsilon)(\|v_i\|^2 + \|v_j\|^2))$$

5) The JL lemma also ensures that for each vector:

$$(1 - \epsilon)\|v_i\|^2 \leq \|f(v_i)\|^2 \leq (1 + \epsilon)\|v_i\|^2$$

6) Using the upper bound from (5):

$$f(v_i) \cdot f(v_j) \leq \frac{1}{2}((1 + \epsilon)(\|v_i\|^2 + \|v_j\|^2) - (1 - \epsilon)(\|v_i\|^2 + \|v_j\|^2))$$

7) Simplifying:

$$f(v_i) \cdot f(v_j) \leq \epsilon(\|v_i\|^2 + \|v_j\|^2)$$

8) This shows that the dot product of any two projected vectors is bounded by a small value proportional to $\epsilon$, which can be made arbitrarily small by increasing $m$.

Therefore, while we cannot have more than $n$ strictly orthogonal vectors in $\mathbb{R}^n$, we can project $k > n$ vectors into $\mathbb{R}^m$ where they are "nearly orthogonal". The dot product of any two projected vectors is bounded by $\epsilon(\|v_i\|^2 + \|v_j\|^2)$, which approaches zero as $\epsilon \to 0$. $\quad\square$

**Corollary A.4.1.** *The projection provided by the Johnson-Lindenstrauss lemma allows for the representation of more classes than the original embedding dimension while maintaining near-orthogonality, effectively addressing the Curse of Orthogonality.*

