# OpenReview forum: "SimO Loss: Anchor-Free Contrastive Loss for Fine-Grained Supervised Contrastive Learning"
_ICLR.cc/2025/Conference — ICLR 2025 Conference Withdrawn Submission_

### Official Review · Reviewer_ifTS · 2024-10-28

**Soundness:** 3
**Presentation:** 3
**Contribution:** 2
**Rating:** 5
**Confidence:** 4

**Summary:**

The paper introduce SimO loss, an anchor-free contrastive loss objective which optimizes both orthogonality and euclidean distance of the embeddings. The authors argue that this is a departure from existing approaches which typically have focused on either orthogonality or euclidean distance. The paper pertain SimO loss to supervised and semi-supervised settings. The paper include visualizations to demonstrate that SimO can find embeddings with clear separations between classes. The authors argue that SimO advances the state-of-the-art in terms of embedding explainability of contrastive learning methods.

**Strengths:**

1. The paper is easy to read.
2. The paper validate SimO by embedding visualizations which clearly shows formation of semantic structure on CIFAR10.

**Weaknesses:**

1. Model performance metrics over 1 epoch of fine-tuning on frozen backbone features is not standard practice. It would be more convincing if you ran this in a standard setting to SimCLR[1] and SupCon[2], and give a proper reference to benchmarks of related work.
2. The 2 year old contrastive method t-SimCNE can give interpretable cluster visualizations with semantic structure [3], unsupervised without class labels and in raw 2d space on both CIFAR10 and CIFAR100.  SimO pertain to supervised settings which limits the applicability.





[1] Ting Chen, Simon Kornblith, Mohammad Norouzi, and Geoffrey Hinton. A simple framework for contrastive learning of visual representations. In International conference on machine learning,pp. 1597–1607. PMLR, 2020a.

[2] Prannay Khosla, Piotr Teterwak, Chen Wang, Aaron Sarna, Yonglong Tian, Phillip Isola, Aaron Maschinot, Ce Liu, and Dilip Krishnan. Supervised contrastive learning. Advances in neural information processing systems, 33:18661–18673, 2020.

[3] Jan Niklas Bohm, Philipp Berens, and Dmitry Kobak. Unsupervised visualization of image datasets using contrastive learning. In International Conference on Learning Representations, 2023.

**Questions:**

See first point weaknesses.

---

### Official Review · Reviewer_ZHLg · 2024-10-29

**Soundness:** 2
**Presentation:** 3
**Contribution:** 2
**Rating:** 3
**Confidence:** 4

**Summary:**

The paper introduces SimO Loss, an anchor-free contrastive learning loss for supervised learning, designed to create class-specific neighborhoods in embedding spaces that maintain intra-class cohesion while achieving inter-class orthogonality. By leveraging Euclidean distance and orthogonality measures, the method aims to address dimensionality collapse and improve interpretability in contrastive learning models.

**Strengths:**

The paper proposed a novel loss that incorporates orthogonality in embedding space to enhance class separation and interpretability. The proposed method shows its effectiveness within a limited CIFAR-10 benchmark.

**Weaknesses:**

While the approach shows potential in the CIFAR-10 dataset, the limited scope and high computational requirements raise questions about its broader applicability.

**Questions:**

Is there a reason why the proposed loss is applied in the supervised contrastive learning domain, given that contrastive loss is predominantly used in self-supervised learning?

---

### Official Review · Reviewer_FvqR · 2024-10-30

**Soundness:** 1
**Presentation:** 3
**Contribution:** 1
**Rating:** 1
**Confidence:** 5

**Summary:**

The paper proposes SimO, a novel loss for supervised contrastive learning. The loss is anchor-free in the sense that all samples in a batch are compared. Distances and dot products are computed between all samples in a batch, such that dissimilar samples are encouraged to have distant, orthogonal representations, while the opposite is encouraged for similar samples. The supervision (class label) is used to determine whether a pair is similar. Results, in the form of both accuracy and dimensionality analysis, are presented for a ResNet-18 trained on CIFAR-10.

**Strengths:**

[S1] The writing of the paper is clear and easy to follow.

[S2] The theorems and lemmas and well-organized and easy to understand.

**Weaknesses:**

[W1] The empirical results are not thorough. No baselines are provided. Only 1 dataset, and 1 backbone are considered. No other methods are compared against. The only quantitative results in the paper are a single line in a single table.

[W2] The empirical results reflect poorly on the proposed method. The single line that is present has accuracies that, for CIFAR-10, are not good.

[W3] I am not convinced that the orthogonality is maximized in practice (Theorem A.2), since the loss term that would maximize orthogonality is entangled with maximizing the distance. In practice, the distance may be sufficiently large that the orthogonality is not so different compared to other works. Detailed analysis and comparison with baselines would have been very helpful to verify that both terms are handled well (orthogonality and distance) in practice.

**Questions:**

Would this outperform any other method for supervised contrastive learning, anchor-free or otherwise? What is the proof?

How are the embeddings learned by SimO different from those learned by other methods, in practice? What evidence can show this?

---

### Official Review · Reviewer_jEmn · 2024-11-03

**Soundness:** 2
**Presentation:** 2
**Contribution:** 2
**Rating:** 3
**Confidence:** 4

**Summary:**

This paper propose a new contrastive learning method, which reduces distance and orthogonality between embeddings of similar inputs and maximizing distance and orthogonality for dissimilar inputs. It conducted experiments on CIFAR-10 to demonstrate the performance of the proposed method. Visualization shows that the method has the ability to balance class separation with intra-class variability.

**Strengths:**

- The properties and advantages of the proposed loss function is supported by theoretical analysis.
- The empirical results demonstrate the potential of the proposed method to balance class separation with intra-class variability.

**Weaknesses:**

My primary concern with this paper is the lack of comparisons with baseline methods in contrastive and representation learning, such as those mentioned in the related work section. Additionally, the paper only reports a 1-epoch fine-tuning accuracy on CIFAR-10, which is not a standard metric in contrastive and representation learning evaluations. These factors make it difficult to compare the proposed method with prior works and to assess whether it offers any improvement over existing methods.

**Questions:**

What is the purpose of evaluating 1-epoch fine-tuning accuracy? What is the highest accuracy that the proposed method can achieve?

---

### Official Review · Reviewer_PhQY · 2024-11-03

**Soundness:** 1
**Presentation:** 1
**Contribution:** 1
**Rating:** 3
**Confidence:** 4

**Summary:**

The paper introduces a novel anchor-free contrastive learning (AFCL) method utilizing a new loss function called Similarity-Orthogonality (SimO) loss. This approach optimizes both the distance and orthogonality of embeddings for better class separation and intra-class cohesion within a semi-metric space. Key contributions include comprehensive theoretical analysis of the embedding space, and visual validations on the CIFAR-10 dataset demonstrating structured and separable class neighborhoods.

**Strengths:**

The paper aims to tackle the issue of dimensionality collapse, which is an important challenge in existing representation learning approaches.

**Weaknesses:**

1. The writing of the work is extremely poor and unengaging. For instance

- The abstract fails to establish a clear motivation or provide adequate background, making it difficult for readers to understand the context and importance of the research.

- Section 3 reads as a collection of definitions and terminology without meaningful connections or explanations.

- In Section 4.1, the definition of the loss function lacks intuition or insights, making it challenging for readers to grasp the underlying rationale.

- Mathematical notations are unclean for instance towards the end of Section 4.2

- Figure titles and extremely small to understand.

- Experiments section is very hard to understand

The paper, as a whole, gives the impression of disjointed content assembled without a coherent progression.

2. Experimental results aren’t rigorous enough

- First, the experiments conducted are only on a small “toy” dataset in the regime of self-supervised learning

- There are no baselines in the paper

3. Most citations of the popular representation learning approaches are missing from the introduction. For example, SimSiam, BYOL, Barlow Twins, etc. Most citations seem to be quite old and outdated.

4. The proposal method is unclear. On a hypersphere, which is a common setting in self-supervised learning approaches, minimizing the Euclidian distance is equivalent to maximizing the cosine similarity between embeddings. Further, it is unclear what $y$ is. The authors mentioned that “y is a binary label for the entire batch”. What does this mean? There are no insights into the approach.

**Questions:**

1. There is a number “1060000” in the titles of Figure 2(a) and Figure 2(b). What does that refer to?

See the Weaknesses section above

---

### Official Review · Reviewer_VqyY · 2024-11-04

**Soundness:** 2
**Presentation:** 2
**Contribution:** 2
**Rating:** 3
**Confidence:** 4

**Summary:**

The paper proposes 1 new metric learning loss SimO for metric learning, which not only considering the Euclidean distance, but also orthogonality (squared dot product). The paper provides proof that SimO loss is a semi-metric and can help mitigate dimensionality collapse.

**Strengths:**

The proposed loss is simple and easy to implement. The paper also provides proofs to show SimO loss is a semi-metric and can help mitigate dimensionality collapse. Paper shows a number of t-SNE figures to showcase the learned embeddings.

**Weaknesses:**

The major concern I have on the paper is for the evaluation.

1. The paper only shows the result of the proposed SimO loss on toy CIFAR10 dataset under the supervised setting. It’s not very clear if the conclusion can be extended to other large-scale datasets or not. For example, adding examples on ImageNet will be helpful.

2. There are no other baseline losses (e.g. cross-entropy) are compared in the evaluation. It’s not very clear how the proposed loss compared to SOTA. It would be helpful to compare to conventional losses such as cross-entropy and triplet

3. The paper claimed that the loss can be used for semi-supervised learning. However, no evaluation has been conducted on semi-supervised learning. For conducting semi-supervised learning experiment, it's helpful to say masking the label of x% of the samples in a supervised dataset say ImageNet,

Some other issues include:

4. At line 193, the formulation of Euclidean distance is wrong. It shouldn't have $\cdot e_j$ to the end.

5. At line 194, it’s not clear why squared dot product can be used to measure the orthogonality, it’s not only determined by the angle between 2 embeddings but also the norm. It would be helpful if the author can provide a clarification why using squared dot product instead of normalized cross product (normalize both vectors to unit-norm before calculating the cross product.)

6. Overall, it’s also not very clear why different categories need to be orthogonal or near orthogonal. For example, if there are only 2 classes, the optimal solution should be opposite, instead of orthogonal. It would be helpful if the author can provide some justification when the number of categories is significantly higher than the dimension of the feature (where orthogonal is not possible).

**Questions:**

I do not suggest to add significantly more experiments in the rebuttal. I'd be happy to discuss how the proposed method can be extend to semi-supervised setting and the orthogonal questions in 5 and 6.

---

### Note · Authors · 2024-11-27

**Comment:**

We would like to thank the reviewers for their feedback and notes on the paper,

we decided to withdraw the paper as we found some improvments that can be made to stabelize the loss and solve the problem we discussed in the paper. There was also some inaccuracies in the proof we had, it turns out that the metrics proposed are pseudo-metrics(o/d) and metric spaces (d/o), instead of semi-metric. And that there is a resemblance between o/d and the inverse-square law

We will keep updating the paper and resubmit it for future conferences.

Thank you very much.
best,

**Withdrawal Confirmation:**

I have read and agree with the venue's withdrawal policy on behalf of myself and my co-authors.